# Benefit of Consolidation Thoracic Radiotherapy in Extensive-Stage Small-Cell Lung Cancer Patients Treated with Immunotherapy: Data from Slovenian Cohort

**DOI:** 10.3390/ijms26083631

**Published:** 2025-04-11

**Authors:** Marina Čakš, Urška Janžič, Tjaša Rutar, Mojca Unk, Ana Demšar, Katja Mohorčič, Nina Turnšek, Erika Matos, Jasna But-Hadžić

**Affiliations:** 1Department of Oncology, University Medical Centre Maribor, 2000 Maribor, Slovenia; marina.caks@ukc-mb.si (M.Č.); trutar@onko-i.si (T.R.); ana.demsar@ukc-mb.si (A.D.); 2Medical Oncology Unit, University Clinic Golnik, 4204 Golnik, Slovenia; urska.janzic@klinika-golnik.si (U.J.); katja.mohorcic@klinika-golnik.si (K.M.); 3Faculty of Medicine, University of Ljubljana, 1000 Ljubljana, Slovenia; munk@onko-i.si (M.U.); nturnsek@onko-i.si (N.T.); ematos@onko-i.si (E.M.); 4Department of Medical Oncology, Institute of Oncology Ljubljana, 1000 Ljubljana, Slovenia; 5Department of Radiotherapy, Institute of Oncology Ljubljana, 1000 Ljubljana, Slovenia

**Keywords:** extensive stage small cell lung cancer, consolidation thoracic radiotherapy, chemoimmunotherapy, immune checkpoint inhibitors, real-world data

## Abstract

Chemoimmunotherapy (CT/IO) with immune checkpoint inhibitors has recently become the standard of care for extensive-stage small cell lung cancer (ES-SCLC). Given the uncertain role of consolidation thoracic radiotherapy (cTRT) in this setting, we conducted a real-world study to evaluate the efficacy and safety of cTRT in ES-SCLC patients receiving first-line CT/IO. We performed a retrospective analysis of ES-SCLC patients treated with first-line CT/IO in Slovenia from December 2019 to June 2024. Patient characteristics, treatment patterns, survival outcomes, and adverse events were analyzed, with subgroup comparisons based on cTRT administration. Among 208 patients (median age: 66 years), median overall survival was 12.1 months (95% CI: 10.6–13.7). cTRT was administered to 46 patients (22.1%), who had fewer metastases. cTRT was associated with improved OS (17.0 vs. 10.8 months; *p* < 0.001) and was an independent OS predictor (HR = 0.58, *p* = 0.035). Grade ≥ 3 adverse events were similar (26.1% vs. 21.3%), though pneumonitis occurred more frequently with cTRT (6.5% vs. 0%, *p* = 0.001). cTRT may improve survival in ES-SCLC patients treated with CT/IO, with no significant increase in toxicity apart from pneumonitis. Further prospective studies are needed.

## 1. Introduction

Small-cell lung cancer (SCLC) is an aggressive neuroendocrine carcinoma that occurs predominantly in current or former smokers and has an extremely poor prognosis. It represents roughly 15% of all lung cancer cases [1,2] and is traditionally divided into extensive (ES) and limited stage (LS), depending on whether the cancer can be encompassed within a single radiation field. Over 70% of SCLC cases are diagnosed at ES [2,3,4].

For more than 30 years, platinum-based chemotherapy, with or without prophylactic cranial irradiation (PCI), remained the standard of care for extensive-stage small-cell lung cancer (ES-SCLC) [5,6,7]. While this approach often achieved high initial objective response rates (ORR), median overall survival (mOS) was limited to around 10 months [8,9]. Consolidation thoracic radiotherapy (cTRT) has been shown in prior meta-analyses and prospective studies to improve both local control (LC) and overall survival (OS) in patients who achieve a response to chemotherapy [10,11,12].

The addition of immunotherapy (IO) with immune checkpoint inhibitors (ICIs) has recently become a new therapeutic option for ES-SCLC. The IMpower133 and CASPIAN randomized trials confirmed that chemoimmunotherapy (CT/IO), a combination of IO with anti-programmed death-ligand 1 (anti-PD-L1) antibodies atezolizumab or durvalumab and four cycles of chemotherapy with platinum/etoposide, resulted in a significant increase in mOS compared to chemotherapy alone, without significantly increasing the incidence of adverse events (AEs) [13,14]. Atezolizumab and durvalumab have received global approval and are now incorporated into the standard first-line treatment for ES-SCLC [15,16]. In Slovenia, both atezolizumab and durvalumab were launched for ES-SCLC at the end of 2019.

cTRT after CT/IO could further improve LC and survival outcomes in selected patients, offering promising potential in the era of IO. However, the feasibility and safety of this approach remain uncertain, as it was not included in the two aforementioned prospective clinical trials [13,14], and only limited data from early-phase clinical trials and retrospective studies are available in the literature [17,18,19,20,21,22,23,24,25,26].

Here, we report the findings of a real-world observational study evaluating the treatment outcomes of patients with ES-SCLC treated with first-line CT/IO in Slovenia. Specifically, we aim to evaluate the efficacy and safety of combining cTRT with CT/IO.

## 2. Results

### 2.1. Overall Study Population (Baseline Characteristics & Overall Survival Analysis)

From December 2019 to June 2024 there were 208 patients with ES-SCLC treated with CT/IO in Slovenia. Baseline characteristics are shown in Table 1. The median age was 66 years (range 41–79 years), 55.3% of patients were 65 years or older and 55.8% of patients were male. The majority of patients were smokers (98.6%). Most patients (74%) were in a good PS of 0–1. Brain metastases were present in 19.7%, liver metastases in 43.8%, and bone metastases in 34.1% at the time of diagnosis and 34.6% of patients had 3 or more metastatic sites involved.

Durvalumab and atezolizumab were administered to 142 (68.3%) and 66 (31.7%) patients, respectively. In the induction phase, patients received platinum-based chemotherapy concurrently with IO, the median number of induction cycles was 4 (range 1–6). The induction phase was followed by the maintenance phase with IO only; the median number of maintenance cycles was 3 (range 0–38).

Consolidation radiotherapy was performed in 46 (22.1%) patients. In all patients, the primary tumor and the affected regional lymph nodes were included in the treatment field. In 7 patients, metastatic sites were also irradiated (suprarenal gland, kidney, and lymph nodes). The median interval between chemoimmunotherapy and cTRT was 42 days (range 14–111), and in more than 90% of patients, cTRT was started within 60 days after the last CT/IO cycle. Radiotherapy (RT) doses ranged from 20 Gy to 60 Gy, with 80.4% (37/46) of patients receiving 30 Gy in 10 once-daily fractions. Other fractionations used were 5 × 4 Gy (1 patient), 13 × 3 Gy (3 patients), 18 × 2.5 Gy (1 patient) and 30 × 2 Gy (4 patients). Prophylactic cranial irradiation (PCI) was delivered at the discretion of the treating physician. Only 3 patients (1.4%) received PCI.

The ORR was 84.9%, with 5.5% and 79.4% of patients achieving a complete response (CR) and partial response (PR), respectively. Progressive disease was observed in 6.5% of patients.

With a median follow-up time of 25.9 months, the mPFS was 7.1 months (95% CI 6.4–7.8; Figure 1) and mOS was 12.1 months (95% CI 10.6–13.7; Figure 2). The estimated 12-month PFS rate was 20%, estimated 12 and 24-month OS rates were 51% and 19%, respectively.

In the univariate Cox regression analysis, several factors were significantly associated with OS. Female sex, median number of CT/IO cycles, PR or CR at evaluation (vs. stable disease), and cTRT were associated with better survival. Conversely, poor ECOG PS (≥2 vs. 0–1), liver metastases, bone metastases, involvement of ≥3 metastatic sites, M1c stage, and high LDH levels at presentation were associated with worse survival (Table 2). Age (≥65 vs. <65), brain metastases (yes vs. no), T stage (Tx-T2 vs. T3-T4), N stage (N0-N2 vs. N3), M1a, M1b stage and type of ICI (durvalumab or atezolizumab) had no influence on OS.

In the multivariate Cox regression analysis, female sex, PR or CR at evaluation, and cTRT remained independently associated with improved survival. Poor ECOG PS (≥2 vs. 0–1) and involvement of ≥3 metastatic sites remained significant predictors of worse OS (Table 2). Liver metastases, bone metastases, M1c stage, high LDH levels, and a median number of CT/IO cycles lost statistical significance after adjustment, suggesting their effects were confounded by other variables.

### 2.2. cTRT vs. Non-cTRT Subgroup Analysis

At diagnosis, there were significant differences in terms of disease burden and treatment parameters between the cTRT group (*n* = 46) and the non-cTRT group (*n* = 162) (Table 3). Patients in the cTRT group had more T3-T4 stage, but were less likely to have liver metastases, bone metastases, more than three metastatic sites involved, and M1c stage. In addition, the median LDH level was lower in the cTRT group.

In terms of treatment exposure, patients in the cTRT group received a higher median number of CT/IO cycles (median 5, range 4–5) compared to the non-cTRT group (median 4, range 4–4). In addition, they received more cycles of maintenance IO (median 5.5, range 2–9 vs. median 2, range 1–4, respectively).

At first progression, the proportion of patients with intrathoracic progression with or without systemic progression was significantly lower in the cTRT group (35.5% vs. 65.9%, respectively). There was no difference in systemic sites of progression between the groups.

Median PFS was 9.7 months in the cTRT group and 6.1 months in the non-cTRT group (HR = 0.44, 95% CI 0.30–0.65, *p* < 0.001; Figure 3) and the estimated 12-month PFS rate was 37% in the cTRT group and 15% in non-cTRT group. Median OS was 17.0 months in the cTRT group and 10.8 months in the non-cTRT group (HR = 0.58, 95% CI 0.35–0.96, *p* = 0.035; Figure 4). The estimated 12 and 24-month OS rates were 79% and 32% in the cTRT group compared with 43% and 16% in the non-cTRT group.

### 2.3. Safety

Out of 208 patients, 47 (22.6%) experienced grade 3 or 4 CTCAE adverse events, the majority of which were related to systemic treatment. Grade 3 or 4 hematological toxicities occurred in 19 patients (9.0%), while 20 patients (9.6%) experienced immune-related adverse events (irAEs) grade 3 or 4. Hepatitis was the most common, occurring in four patients.

No significant differences in the incidence of grade 3 or 4 AEs or irAEs between the cTRT and non-cTRT groups (26.1% vs. 21.3% and 8.7% vs. 9%, respectively) were observed. Pneumonitis grade 3 or 4 was reported in three (1.4%) patients, all of whom had received thoracic RT during systemic treatment. The difference in pneumonitis in cTRT vs. non-cTRT group was significant (6.5% vs. 0%, respectively; *p* = 0.001) (Table 4). No cases of grade 5 pneumonitis were reported.

## 3. Patients and Methods

The study included treatment-naïve consecutive patients with pathologically confirmed ES-SCLC treated with first-line CT/IO between December 2019 and June 2024 in routine clinical practice at three academic institutions in Slovenia: the Institute of Oncology Ljubljana, University Clinic Golnik, and University Clinical Center Maribor, in which all lung cancer patients in the country are treated. The study was conducted in accordance with the Declaration of Helsinki and was approved by the Ethics Committee and Review Board of the Institute of Oncology Ljubljana (No. ERIDNPVO-0076/2024). Individual patient consent was waived based on the retrospective design of the study, the minimal risk to participants, and the fact that the institution’s standard treatment consent forms already included permission for the use of patient data, biological materials, and test results for research purposes.

Patients received at least one cycle of CT/IO with the anti-PD-L1 antibody atezolizumab or durvalumab, followed by maintenance therapy of the anti-PD-L1 antibody until disease progression or unacceptable toxicity. Patients treated for LS-SCLC and those not receiving IO were excluded. Consolidation of thoracic radiotherapy after CT/IO and PCI was allowed and decided by the multidisciplinary tumor board. Only patients without progressive disease after CT/IO were eligible for cTRT.

Patients’ clinical characteristics were taken from the medical records by the treating oncologists. The data collected included age, sex, smoking status, Eastern Cooperative Oncology Group performance status (ECOG PS), metastatic sites, tumor stage, laboratory results at diagnosis, and treatment patterns. Response to treatment was evaluated by computer tomography (CT) scan, according to the Response Evaluation Criteria for Solid Tumors (RECIST) version 1.1. Adverse events were assessed and graded based on the Common Terminology Criteria for Adverse Events (CTCAE) v 5.0 [27].

Statistical analyses were conducted using SPSS (version 26.0; IBM Corp., Armonk, NY, USA). Descriptive statistics were used to describe patient characteristics and AEs. OS was defined as the time from diagnosis until death from any cause or the last follow-up visit. Progression-free survival (PFS) was defined as the time from diagnosis until the time of disease progression, death from any cause, or last follow-up visit. Survival analysis was performed using the Kaplan–Meier method. Differences in survival between the cTRT and non-cTRT groups were analyzed using the log-rank test and the Cox proportional hazards regression model. A *p*-value of less than 0.05 was considered statistically significant. The chi-square test and non-parametric tests were used to compare variables between the cTRT and non-cTRT groups.

## 4. Discussion

The present study provides results from an observational study evaluating the real-world efficacy of first-line CT/IO in patients with ES-SCLC. The median OS of patients in our cohort (12.1 months) was comparable to those observed in the pivotal trials (12.3 months in Impower 133, 13.0 months in CASPIAN) [13,14], despite the fact that our study included a more heterogeneous real-world population with a higher proportion of patients with worse PS, more advanced age and involvement of brain metastases, which are known poor prognostic factors in patients with ES-SCLC. Similar conclusions were made in other published real-world prospective and retrospective studies [28,29,30,31,32,33,34].

Our subgroup analysis suggests that the addition of cTRT to CT/IO may improve survival. The role of cTRT in ES-SCLC was established long before the introduction of IO. In 1999, Jeremic et al. first demonstrated that the addition of cTRT to platinum-based CT significantly improved OS [11]. The CREST trial in 2015 further validated its efficacy by enrolling 495 patients with ES-SCLC who had shown response to CT. The patients in the cTRT group were treated with thoracic irradiation of 30 Gy in 10 fractions. The 1-year and 2-year OS rates were higher in the cTRT group compared to the control group (33% vs. 28%, *p* = 0.066; and 13% vs. 3%, *p* = 0.004, respectively) [12]. The RTOG 0937 study reported delayed progression with cTRT but did not show a significant improvement in 1-year OS. However, no difference in survival was observed between early and late cTRT administration [35]. A meta-analysis later confirmed that the addition of cTRT to first-line platinum-based CT reduced disease progression and improved OS [10]. Findings from this meta-analysis led to the incorporation of cTRT into the standard of care for patients with ES-SCLC in response to the first-line CT, though practices globally varied as there was no clear consensus on the application of cTRT [36,37].

With the introduction of CT/IO as the new standard of care, the role of cTRT has been further debated. The IMpower133 and CASPIAN trials did not include cTRT, as there was limited safety data on combining thoracic RT with IO [13,14]. However, the benefit of the addition of IO to CT remains modest, and the most common site of progression after first-line CT/IO is locoregional [38]. In addition, preclinical and clinical studies suggest a potential synergy between radiotherapy and immunotherapy, supporting the rationale for further investigation the putative beneficial role of cTRT in the CT/IO era [39,40,41].

Welsh et al. confirmed the safety and promising efficacy of combining pembrolizumab with TRT after induction CT in a phase 1 trial, providing a rationale for this synergistic therapeutic approach [17]. Subsequently, several retrospective studies have demonstrated improved OS in patients receiving both CT/IO and cTRT [18,19,20,21,22,23,25,26]. A recent meta-analysis, which included 12 retrospective and 3 prospective studies, suggested that integrating cTRT with CT/IO improves survival outcomes [24]. Ongoing randomized clinical trials are currently evaluating the efficacy and safety of such an approach (NCT05223647, NCT04462276, and NCT04402788).

In our analysis, patients who received cTRT had a significantly longer mOS compared to those who did not receive cTRT (17.0 months vs. 10.8 months, *p* < 0.001). After adjustment for confounders, cTRT remained an independent predictor of improved survival (HR = 0.58, 95% CI 0.35–0.96, *p* = 0.035). However, the possibility of selection bias should be considered, as the cTRT group had a more favorable prognosis, including fewer liver and bone metastases and a lower overall metastatic burden, and additionally, patients in this group also received more cycles of systemic therapy. These imbalances suggest that the observed survival benefit may be partially influenced by baseline differences rather than cTRT alone. However, at least three retrospective studies (Xie et al., Peng et al., and Yao et al. [19,21,25]) evaluated cTRT in combination with CT/IO with a similar or greater number of patients (45, 57, and 99, respectively). In these studies, the cTRT and non-cTRT groups were better balanced, with Peng et al. and Yao et al. utilizing propensity score matching to minimize selection bias. Notably, all studies included more patients with liver metastases in the cTRT groups (31% in Xie et al., 25% in Peng et al., and 24% in Yao et al.). Nevertheless, all studies reported improved progression-free survival (PFS) and OS with cTRT, suggesting that the survival benefit observed in our study is unlikely to be solely due to group imbalances [19,21,25]. Still, our survival benefit observed in the cTRT group must be interpreted with caution due to group imbalance.

The ideal cTRT dose in the context of immunotherapy remains uncertain due to the absence of prospective studies. Additionally, the sequencing of cTRT and IO maintenance therapy has not been well-defined.

In our cohort, 80.4% of patients received radiotherapy with 30 Gy in 10 once-daily fractions. cTRT was initiated after the completion of CT/IO, during the IO maintenance phase. The majority of patients began cTRT within 60 days following their last cycle of CT/IO.

A retrospective study found no significant difference in outcomes between early cTRT (administered within ≤3 cycles of chemotherapy) and late cTRT (administered after >3 cycles of chemotherapy) [42]. Han et al. suggested that initiating cTRT within 6 cycles of chemotherapy may improve local control [43]. In a study by Peng et al., patients with ES-SCLC received cTRT primarily during IO maintenance, while 15 patients (26.3%) underwent synchronous cTRT within the first two cycles of systemic therapy [21]. However, safety and efficacy data were not reported. Several prospective trials are currently investigating the role of cTRT, with most studies administering cTRT during maintenance IO. Until these results are available, the American Society of Clinical Oncology (ASCO) guidelines recommend that cTRT should be administered within 6–8 weeks after CT/IO and before the start of IO maintenance with 30 Gy delivered in 10 fractions [37]. This regimen was employed in both the CREST trial and a phase 1/2 study evaluating ipilimumab and nivolumab in ES-SCLC, with no significant toxicity reported in either study [12,44]. Higher doses (45 Gy in 15 fractions) have been tested in RTOG 0937 and a phase 1 trial of pembrolizumab [17,35]. Several retrospective studies have employed different cTRT protocols, but no direct comparisons between them have been made. However, evidence suggests that high-dose cTRT may improve LC and OS in ES-SCLC while potentially leading to an increased risk of pulmonary toxicity [18,45].

In our analysis, the treatment regimen was well tolerated, with a manageable incidence of grade 3 or higher AEs. Overall, there were no significant differences between the cTRT and non-cTRT groups. All cases of pneumonitis occurred in the cTRT group; however, the incidence was low, and all cases were effectively managed.

IO combined concurrently or sequentially with TRT, may increase pulmonary toxicity. Following chemoradiotherapy, the rate of grade 2 or higher pneumonitis was reported in up to 30% [46]. The incidence of immune-related pneumonitis after IO ranges from 3 to 5% [47,48]. Treatment with IO after conventional TRT resulted in up to 62% of any grade and 1–9% of grade 3 pneumonitis [49]. Safety data for the combination of IO and TRT in ES-SCLC patients have also been published. In a phase 1 study with 38 patients, concurrent thoracic radiotherapy and pembrolizumab was well tolerated. No grade 4 or 5 toxicities were observed, and grade 3 toxicity occurred in only 6% of patients [17]. In real-world studies, the incidence of grade 3 pulmonary toxicity was approximately 0–9% [18,19,20]. In addition, a meta-analysis confirmed a manageable incidence of grade 3 AEs and radiation pneumonitis [24]. Our safety results are consistent with published data and overall, the toxicity of combining TRT with immunotherapy appears to be manageable.

This study provides valuable insights into the efficacy of CT/IO in ES-SCLC patients in a real-world setting, where the patient population differs from that in clinical trials. It also offers real-world data on the efficacy and safety of cTRT in the immunotherapy era, an area where prospective data are lacking. Additionally, most retrospective studies have been conducted in Asian and North American populations, with only limited data available from European countries. As patient characteristics, healthcare systems, access to drugs, and healthcare facilities may differ between regions, findings from North America and Asia may not be fully generalizable to the Central European population. We acknowledge the limitations of this study. As a retrospective analysis of a real-world population, it is prone to selection bias. The cTRT group was relatively small, and the observed OS benefit may partly reflect a more favorable prognosis of selected patients. Due to the small cohort size, propensity score matching to minimize selection bias was not performed. In addition, we could only report grade ≥ 3 toxicities due to the retrospective study design. Furthermore, there is a tendency of the underreporting AE in real-world clinical settings [31]. Additionally, the distinction between irAE pneumonitis and radiation pneumonitis is challenging, as their clinical presentations overlap, and their management strategies are similar. We acknowledge the low proportion of patients receiving PCI in our cohort. As our focus was on cTRT with CT/IO, PCI was not explored in detail. Its limited use reflects current practice in Slovenia, influenced by emerging evidence, uncertainties in the IO era, and routine CNS imaging.

## 5. Conclusions

This analysis suggests that the addition of cTRT to first-line CT/IO may be associated with prolonged OS and a manageable safety profile. However, prospective clinical studies are needed to confirm these findings. Furthermore, determining the optimal type, dose, and timing of TRT in combination with IO is crucial for improving patient outcomes, underscoring the need for further research.

## Figures and Tables

**Figure 1 ijms-26-03631-f001:**
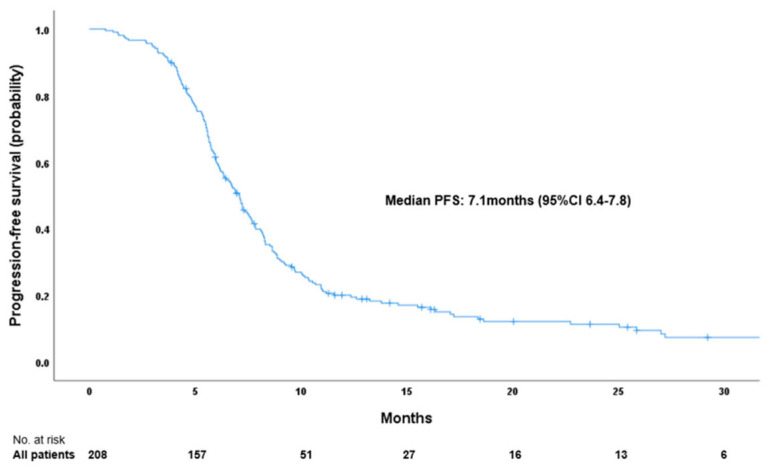
Kaplan–Meier curve showing progression-free survival for all 208 enrolled patients with ES-SCLC.

**Figure 2 ijms-26-03631-f002:**
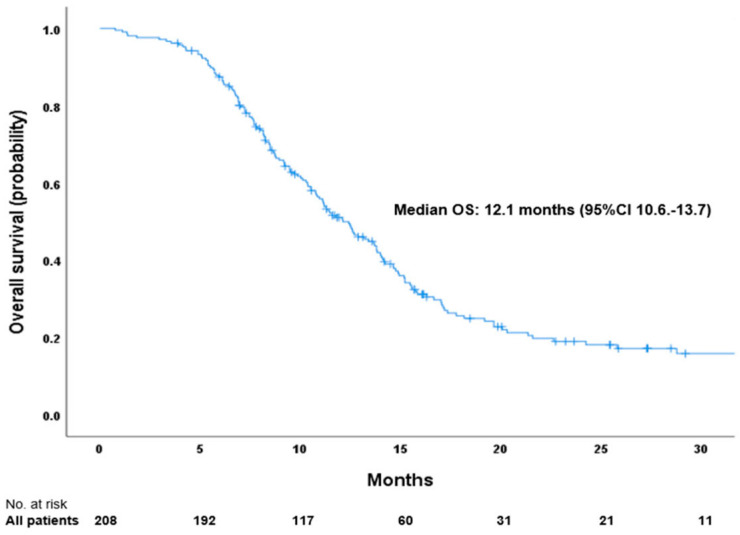
Kaplan–Meier curve showing overall survival for all 208 enrolled patients with ES-SCLC.

**Figure 3 ijms-26-03631-f003:**
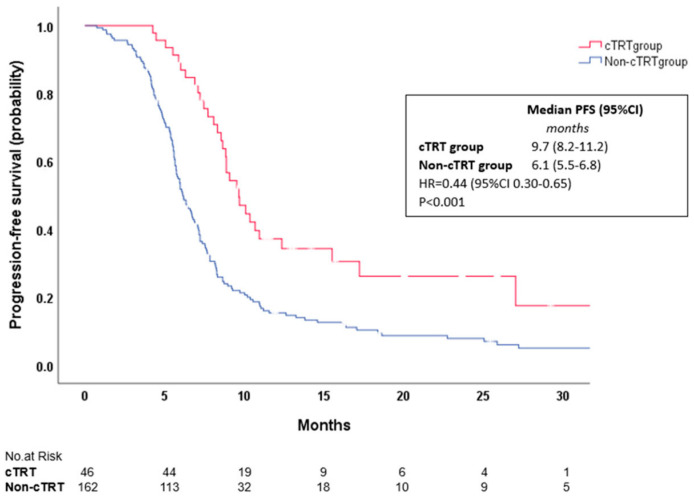
Kaplan–Meier curves showing progression–free survival for ES–SCLC patients in two treatment groups.

**Figure 4 ijms-26-03631-f004:**
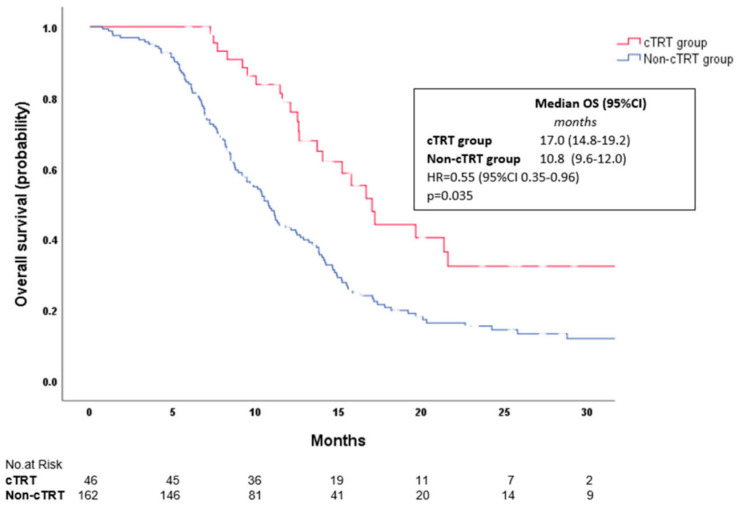
Kaplan–Meier survival curves of overall survival of ES–SCLC patients in two treatment groups.

**Table 1 ijms-26-03631-t001:** Patients and disease characteristics of 208 included patients.

Characteristics at Diagnosis	All Included Patients (N = 208)
Median Age (years)	66 (range 41–79)
Age ≥ 65 y (%)	55.3
Male Patients (%)	55.8
Smokers (former/current) (%)	98.6
ECOG PS 0–1 (%)	74
Brain Metastasis (%)	19.7
Liver Metastasis (%)	43.8
Bone Metastasis (%)	34.1
Patients with ≥3 Metastatic Sites (%)	34.6
T stage Tx–T2 (%) T3–T4 (%)	26.4 73.6
N stage N0–N2 (%) N3 (%)	39.9 60.1
M stage M1a (%) M1b (%) M1c (%)	9.6 9.6 78.4
Median LDH	4.29 (95% CI 3.98; 4.79)
Patients with Elevated LDH (%)	53.4

ECOG = Eastern Cooperative Oncology Group; PS = performance status; LDH = lactate dehydrogenase.

**Table 2 ijms-26-03631-t002:** Univariate and multivariate Cox regression analyses of overall survival (n = 208).

Variable	Univariate HR (95% CI)	*p*-Value	Multivariate HR (95% CI)	*p*-Value
Sex (female vs. male)	0.67 (0.49–0.93)	0.017	**0.59 (0.40–0.86)**	**0.007**
ECOG PS (≥2 vs. 0–1)	1.76 (1.23–2.52)	0.002	**2.24 (1.43–3.50)**	**<0.001**
Liver Metastasis (yes vs. no)	1.68 (1.22–2.32)	0.001	1.20 (0.79–1.80)	0.393
Bone Metastasis (yes vs. no)	1.55 (1.12–2.16)	0.009	0.98 (0.64–1.49)	0.916
Metastatic sites (≥3 vs. 0–2)	2.09 (1.50–2.92)	<0.001	**1.97 (1.28–3.03)**	**0.002**
M stage M1a vs. M1c M1b vs. M1c	0.64 (0.037–1.11) 0.42 (0.21–0.83)	0.018 0.116 0.013	1.29 (0.68–2.49) 0.57 (0.24–1.36)	0.281 0.447 0.202
LDH (high vs. normal)	1.76 (1.27–2.44)	0.001	1.41 (0.94–2.09)	0.095
Response evaluation (CR/PR vs. SD)	0.49 (0.27–0.87)	0.014	**0.37 (0.20–0.71)**	**0.003**
CT/IO Cycles (median)	0.79 (0.64–0.97)	0.026	0.99 (0.80–1.23)	0.936
cTRT (yes vs. no)	0.44 (0.28–0.68)	<0.001	**0.58 (0.35–0.96)**	**0.035**

ECOG = Eastern Cooperative Oncology Group; PS = performance status; LDH = lactate dehydrogenase; CR = complete response; PR = partial response; SD = stable disease; CT/IO = chemoimmunotherapy, cTRT = consolidation thoracic radiotherapy; HR = hazard ratio; CI = confidence interval. Values in bold indicate statistical significance in the multivariate analyses (*p* < 0.05).

**Table 3 ijms-26-03631-t003:** Patient and treatment characteristics: Chemoimmunotherapy with or without consolidation radiotherapy.

Characteristics at Diagnosis	cTRT Group (*n* = 46)	Non-cTRT Group (*n* = 162)	*p*-Value
Median age (years)	66 (range 61–70)	66 (range 61–69)	0.834
Age ≥ 65 years (%)	56.5	54.9	0.849
Male Sex (%)	56.5	55.6	0.907
ECOG PS 0–1 (%)	80.4	72.8	0.459
Brain Metastasis (%)	19.6	19.8	0.977
Liver Metastasis (%)	13.0	52.5	**<0.001**
Bone Metastasis (%)	21.7	37.7	**0.045**
Patients with ≥3 Metastatic sites (%)	19.6	38.9	**0.015**
T stage Tx–T2 (%) T3–T4 (%)	6.5 95.3	32.1 67.9	**0.001**
N stage N0–N2 (%) N3 (%)	37.0 63.0	40.7 59.3	0.644
M stage M1a (%) M1b (%) M1c (%)	21.7 17.4 56.5	6.2 7.4 84.6	**<0.001**
Median LDH	3.81 (95%CI 3.50;4.29)	4.67 (95%CI 4.12;5.37)	**0.005**
Patients with elevated LDH (%)	41.3	56.8	0.063
Median No. Of CT/IO Cycles (n)	5 (range 4–5)	4 (range 4–4)	**0.049**
Type of ICI Durvalumab (%) Atezolizumab (%)	56.5 43.5	71.6 28.4	**0.052**
Response before cTRT CR/PR SD	93.5 6.5	82.4 9.2	**0.478**
Median No. Of Maintenance IO cycles (n)	5.5 (range 2–9)	2 (range 1–4)	**<0.001**
Intrathoracic +/− systemic Progression (%)	35.5	65.9	**0.002**

cTRT = consolidation thoracic radiotherapy; ECOG = Eastern Cooperative Oncology Group; PS = performance status; No. = number, CT/IO = chemoimmunotherapy; IO = immunotherapy; ICI = immune checkpoint inhibitor; CR = complete response, PR = partial response; SD = stable disease. Values in bold indicate statistically significant differences between the cTRT and non-cTRT groups (*p* < 0.05).

**Table 4 ijms-26-03631-t004:** Grade 3 and 4 adverse events in two treatment groups of ES-SCLC patients.

Adverse Event G3/4	cTRT Group (*n* = 46)	Non-cTRT Group (*n* = 160)
All	12 (26.1%)	34 (21.3%)
Neutropenia	2 (4.3%)	10 (6.3%)
Anemia	1 (2.2%)	5 (3.1%)
Thrombocytopenia	1 (2.2%)	4 (2.5%)
irAE	4 (8.7%)	15 (9.0%)
Pneumonitis	3 (6.5%)	0 (0%)
Esophagitis	1 (2.2%)	0 (0%)

cTRT = consolidation thoracic radiotherapy; irAE = immune-related adverse event.

## Data Availability

The data presented in this study are available on request to the corresponding author due to ethical reasons.

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
