# Peer review of "Benefit of Consolidation Thoracic Radiotherapy in Extensive-Stage Small-Cell Lung Cancer Patients Treated with Immunotherapy: Data from Slovenian Cohort"

_ijms, 2025, doi:10.3390/ijms26083631_

Round 1
Reviewer 1 Report
Comments and Suggestions for Authors
This article investigates very important topic, since it is well known that patients with ES small cell lung cancer have poor survival, despite the significant advances in the treatment in the past decade.
I suggest that authors make some correction since the iThenticate report showed 33% match.
It would be interesting if authors could add the sites of progression for those two groups. It would be also interesting to add outcomes in patients with brain and liver metastases in general and by groups since it is well known that those patients have worse prognosis compared to patients who do not have this sites affected.
Author Response
Dear Reviewer 1,
Thank you for taking the time to review our article. We have carefully revised the manuscript in accordance with your suggestions. All modifications have been highlighted in the revised version for your convenience.
I suggest that authors make some correction since the iThenticate report showed 33% match.
We analysed the text with Turnitin and made corrections where there was a match. Some of the text can not be corrected, since it is well established and standard wording. For example “ECOG = Eastern Cooperative Oncology Group; PS= performance status; LDH=lactate dehydrogenase” or “Kaplan-Meier survival curve of overall survival”. The corrected text due to matching is done in blue.
It would be interesting if authors could add the sites of progression for those two groups. It would be also interesting to add outcomes in patients with brain and liver metastases in general and by groups since it is well known that those patients have worse prognosis compared to patients who do not have this sites affected.
Thank you for your comment. We have reviewed the data on sites of progression. The only observed difference between the subgroups was in local progression, while no significant differences were found in other sites of progression, including the liver, bone, brain, and other collected sites. We have incorporated this statement into the text. 164-165.
In our overall population, patients with liver metastases had worse survival than those without, as observed in the univariable analysis (UVA) for OS. However, no significant difference in OS was found between patients with and without brain metastases. Similarly, when comparing OS between subgroups, no significant differences were observed.
Reviewer 2 Report
Comments and Suggestions for Authors
Summary:
This study offers timely real-world evidence regarding the role of consolidation thoracic radiotherapy (cTRT) in patients with extensive-stage small-cell lung cancer (ES-SCLC) who have received first-line chemoimmunotherapy (CT/IO). The authors report a significant survival benefit associated with cTRT, as evidenced by a median overall survival (OS) of 17.0 months compared to 10.8 months in the control group (HR=0.51). These findings are consistent with emerging retrospective data and address a critical gap in the era of immunotherapy. By contextualizing cTRT within contemporary CT/IO regimens, this work provides valuable insights that can guide clinical decision-making.
Major comments:
1.It is suggested that the baseline data of the cTRT group and the non-cTRT group be supplemented in Table 1. Additionally, it is recommended to add more baseline information, such as T stage, N stage, M stage, type of ICIs, and response evaluation before TRT. These factors should also be included in the Cox regression analysis.
- It is advisable to incorporate PFS data into the survival analysis results.
- It is recommended that the results section include a supplementary table detailing the toxic effects.
- The cTRT group exhibited fewer metastases, a lower metastatic burden, and received a greater number of maintenance immunotherapy cycles, highlighting potential baseline prognostic imbalances. The observed survival benefits might be attributable to patient selection rather than the efficacy of cTRT, thereby challenging causal inferences. Notably, a significant imbalance at baseline was identified between the two patient groups, prompting the recommendation of propensity score matching as a method to reduce selection bias.
- No data is available regarding the reasons for cTRT selection, such as physician preference or response to chemotherapy. Unaccounted factors may influence outcomes, potentially biasing the observed association.
- It is recommended to integrate the latest references from the past year specifically addressing extensive-stage small-cell lung cancer and thoracic radiotherapy.
Minor comments:
7. In line 21 (median age: 66), it is recommended to revise the phrasing to (median age: 66 years) for clarity and consistency in unit representation.
8.The result section states, "In terms of treatment exposure, patients in the cTRT group received a higher median number of CT/IO cycles and more cycles of maintenance IO compared to the non-cTRT group. ", please add relevant specific data here.
9.The result section states, "At first progression, the proportion of patients with intrathoracic progression with or without systemic progression was significantly lower in the cTRT group. ", please add relevant specific data here.
10.Minor grammatical errors, such as "recieved" in line 219, necessitate a thorough review and revision.
Author Response
Dear Reviewer 2,
Thank you for taking the time to review our manuscript. We sincerely appreciate your valuable comments and suggestions, which have helped us improve the quality of our work. All changes made in response to your feedback are highlighted in the revised text for your convenience
Major comments:
1.It is suggested that the baseline data of the cTRT group and the non-cTRT group be supplemented in Table 1. Additionally, it is recommended to add more baseline information, such as T stage, N stage, M stage, type of ICIs, and response evaluation before TRT. These factors should also be included in the Cox regression analysis.
Thank you for your comment. In Table 1, we have presented the characteristics of the total population for which an overall survival analysis was performed to facilitate comparison with the pivotal studies. The baseline data for both groups (cTRT and non-cTRT) are presented in Table 3. We agree that information on TNM in the two groups (cTRT and non-cTRT) would be very valuable, but unfortunately we did not collect these data from the medical records as they were not written in a consistent manner and much data would be missing. We added the type of ICI and response to CT/IO as suggested. 135-136, 139, 141, 144-145, 147( Table 2)
- It is advisable to incorporate PFS data into the survival analysis results.
We analyzed progression-free survival (PFS) and included it in the results. 91-93, 166-168,177 However, we did not perform univariable (UVA) or multivariable analyses (MVA) for PFS, as it would significantly increase the volume of the article. Moreover, given the retrospective nature of the study, PFS is heavily influenced by the timing and frequency of diagnostic evaluations.
- It is recommended that the results section include a supplementary table detailing the toxic effects.
Thank you for the valuable suggestion. We have added Table 4, which provides a detailed overview of the reported adverse events.198,210-211
- The cTRT group exhibited fewer metastases, a lower metastatic burden, and received a greater number of maintenance immunotherapy cycles, highlighting potential baseline prognostic imbalances. The observed survival benefits might be attributable to patient selection rather than the efficacy of cTRT, thereby challenging causal inferences. Notably, a significant imbalance at baseline was identified between the two patient groups, prompting the recommendation of propensity score matching as a method to reduce selection bias.
Thank you for the important observation. We agree that the survival advantage observed in the cTRT group may be influenced by prognostic imbalances between the groups. We added the number of CT/IO into MVA survuval analysis and it was not significant.The issue of group imbalance has been thoroughly addressed in the discussion and acknowledged as a limitation of the study. Due to the small number of patients in the cTRT group and the clear selection of patients for cTRT by the multidisciplinary team, we did not perform propensity score matching. This rationale has been included in the limitations section of the study. 321-323
- No data is available regarding the reasons for cTRT selection, such as physician preference or response to chemotherapy. Unaccounted factors may influence outcomes, potentially biasing the observed association.
We agree that there was a selection bias in the decision-making process for cTRT within the multidisciplinary team or treating physician. Unfortunately, we do not have detailed data on the specific factors that influenced patient selection. However, we have clarified in the text that the only formal criterion for receiving cTRT was the absence of disease progression following chemotherapy and immunotherapy.80-81
- It is recommended to integrate the latest references from the past year specifically addressing extensive-stage small-cell lung cancer and thoracic radiotherapy.
We agree and have revised the literature accordingly. We added Ref 25 and 26 to support and strengthen the discussion
Minor comments:
7. In line 21 (median age: 66), it is recommended to revise the phrasing to (median age: 66 years) for clarity and consistency in unit representation.
Corrected, thank you.
8.The result section states, "In terms of treatment exposure, patients in the cTRT group received a higher median number of CT/IO cycles and more cycles of maintenance IO compared to the non-cTRT group. ", please add relevant specific data here.
Added, thank you.
9.The result section states, "At first progression, the proportion of patients with intrathoracic progression with or without systemic progression was significantly lower in the cTRT group. ", please add relevant specific data here.
Added, thank you.
10.Minor grammatical errors, such as "recieved" in line 219, necessitate a thorough review and revision.
Thank you for observation. We gramatically reviewed the revision.
Reviewer 3 Report
Comments and Suggestions for Authors
Major point.
1) Many studies have already reported that adding cTRT after chemo-immunotherapy has a survival benefit. How does this study differ from previous studies? What is the real purpose of this study? Was this study conducted to confirm the results of previous studies by conducting analyses similar to those of many previously published studies?
I did not find any unique features of this study that were different from previous studies. Authors should describe the originality of their study in more detail in the Introduction and Discussion sections.
2) The cTRT group and the non-cTRT group are very different. The difference between the two groups has a very important impact on the results. However, the authors describe it with the nuance that the results of this study and previous studies are similar, so the difference between the two groups is not a big problem. A more convincing explanation is needed.
Minor point
Why was PCI performed on only 1.3% of all patients? 84.9% of patients responded to chemo-immunotherapy, but the proportion of patients who underwent PCI was too low. Please state a clear reason.
Author Response
Dear Reviewer 3,
Thank you for taking the time to review our manuscript. We greatly appreciate your insightful comments and constructive feedback. We have addressed your suggestions in the revised version of the article, and all changes have been highlighted accordingly.
1) Many studies have already reported that adding cTRT after chemo-immunotherapy has a survival benefit. How does this study differ from previous studies? What is the real purpose of this study? Was this study conducted to confirm the results of previous studies by conducting analyses similar to those of many previously published studies?
I did not find any unique features of this study that were different from previous studies. Authors should describe the originality of their study in more detail in the Introduction and Discussion sections.
Thank you for this important comment and for the opportunity to clarify the purpose and originality of our study. The primary aim of this study was not merely to replicate existing data, but to confirm and expand upon prior findings by providing additional real-world evidence on the efficacy and safety of consolidation thoracic radiotherapy (cTRT) following chemo-immunotherapy (CT/IO), specifically within a national healthcare system characterized by centralized oncology care. Importantly, this study also contributes valuable data on a European population, which remains underrepresented in the current literature.
In response to your comment, we have added a section addressing this point in the Discussion to better highlight the originality and relevance of our work.311-318
2) The cTRT group and the non-cTRT group are very different. The difference between the two groups has a very important impact on the results. However, the authors describe it with the nuance that the results of this study and previous studies are similar, so the difference between the two groups is not a big problem. A more convincing explanation is needed.
Thank you for this valuable comment. We fully acknowledge that the groups in our study are not directly comparable, and that patient selection for cTRT was influenced by clinical judgment within a multidisciplinary team, leading to inherent selection bias. Our intention was not to minimize the significance of this imbalance but rather to place our findings within the context of existing literature, where similar trends have been observed despite methodological differences. That said, we agree that a more detailed and transparent explanation is necessary.To address this, we have revised the Discussion section to clearly state that:
The survival benefit observed in the cTRT group must be interpreted with caution due to potential confounding factors and that propensity score matching was not feasible due to the small sample size, but we have acknowledged this as a limitation of the study. 268-269, 321-323. We corrected the statement in the conclusion 332.
We hope these clarifications provide a more balanced and convincing interpretation of the data. Thank you again for your thoughtful feedback
Minor point
Why was PCI performed on only 1.3% of all patients? 84.9% of patients responded to chemo-immunotherapy, but the proportion of patients who underwent PCI was too low. Please state a clear reason.
As the primary focus of our analysis was to evaluate the efficacy of cTRT with CT/IO, we did not discuss the role of PCI in detail. We acknowledge that the number of patients who received PCI in our study was small. We believe this is partly due to decline in PCI usage over time, initially influenced by the study by Takanashi et al. and further by the uncertain role of PCI in the IO era. In our country, the decision to administer PCI is made by the multidisciplinary team (MDT) and finalized through a discussion between the radiation oncologist and the patient. Since we have »integrated« routine CNS screening (with at least a CT scan, and more often with MRI) into our practice, many patients opt not to receive PCI. We added the explanation in the discussion. 327-330
Round 2
Reviewer 1 Report
Comments and Suggestions for Authors
No further comments
Author Response
We thank Reviewer 1 for their time and thoughtful evaluation. We appreciate that there are no further comments and are grateful for their support of our work.
Reviewer 2 Report
Comments and Suggestions for Authors
- Since the authors acknowledge the importance of TNM staging for prognosis, it is suggested that they supplement this data. Additionally, retrieving and reviewing the original impact images is recommended to ensure accurate staging. These factors should also be included in the Cox regression analysis.
- It is suggested that Figure 1 supplement the PFSfor all 208 enrolled patients with ES-SCLC.
- Since the authors are already aware that the baseline imbalance between the two groups in this study has a significant impact on survival outcomes, I strongly recommend supplementing the analysis with propensity score matching (PSM). The study comprises 46 participants in the cTRTgroup and 162 in the Non-cTRT group. In the context of contemporary research evaluating the role of thoracic radiotherapy in extensive-stage small cell lung cancer, the sample size of this study is not insubstantial. It is therefore suggested that the authors refine the PSM analysis to more accurately delineate the clinical value of thoracic radiotherapy.
- It is recommended to collect additional data on radiation esophagitis for a more comprehensive assessment of toxicity.
Author Response
Dear riewer 2,
thank you again for your valuable suggestions. We believe they have significantly improved our work, though we acknowledge that some limitations remain. We carefully considered all your comments and have incorporated them into our manuscript as thoroughly as possible.
- Since the authors acknowledge the importance of TNM staging for prognosis, it is suggested that they supplement this data. Additionally, retrieving and reviewing the original impact images is recommended to ensure accurate staging. These factors should also be included in the Cox regression analysis.
We reviewed the patient records again and retrieved TNM staging data form initial diagnostic images. This staging data has now been incorporated into our Cox regression analysis. Table 1; 142, 144, 145, 150. Table 2; 161, 162
- It is suggested that Figure 1 supplement the PFS for all 208 enrolled patients with ES-SCLC.
Thank you for suggestion. It was added. Figure 1
- Since the authors are already aware that the baseline imbalance between the two groups in this study has a significant impact on survival outcomes, I strongly recommend supplementing the analysis with propensity score matching (PSM). The study comprises 46 participants in the cTRTgroup and 162 in the Non-cTRT group. In the context of contemporary research evaluating the role of thoracic radiotherapy in extensive-stage small cell lung cancer, the sample size of this study is not insubstantial. It is therefore suggested that the authors refine the PSM analysis to more accurately delineate the clinical value of thoracic radiotherapy.
As per your suggestion, we implemented propensity score matching (PSM) using SPSS to balance the baseline characteristics between treatment groups. However, this approach substantially reduced the size of the control group, resulting in only 36 matched controls for the 46 patients in the cTRT group. This considerable reduction limited our statistical power and undermined the reliability of any conclusions drawn from the matched analysis. Given these constraints, we opted to adjust for multiple key covariates using a multivariable Cox proportional hazards model, which allowed us to account for confounding factors while maintaining the full sample size. We acknowledge that larger, prospective studies would be better positioned to evaluate the impact of cTRT with more robust matching and statistical power.
- It is recommended to collect additional data on radiation esophagitis for a more comprehensive assessment of toxicity
Thank you again for your valuable suggestion. We reviewed our records and have now included
G3 esophagitis in the toxicity table. Table 4
Reviewer 3 Report
Comments and Suggestions for Authors
The authors revised the paper well as I requested.
The quality of the paper has improved a lot by describing the differences between previously reported studies and this paper. The originality of this study is well described in the revised paper.
Author Response
We thank Reviewer 1 for their time and thoughtful evaluation. We appreciate that there are no further comments and are grateful for their support of our work